# Study on Stator-Rotor Misalignment in Modular Permanent Magnet Synchronous Machines with Different Slot/Pole Combinations

Danilo Riquelme [1,*], Carlos Madariaga [2], Werner Jara [1], Gerd Bramerdorfer [3], Juan A. Tapia [2] and Javier Riedemann [4]

[1] School of Electrical Engineering, Pontificia Universidad Católica de Valparaíso, Valparaíso 2362804, Chile
[2] Department of Electrical Engineering, University of Concepcion, Concepcion 4070386, Chile
[3] Department of Electrical Drives and Power Electronics, Johannes Kepler University Linz, 4040 Linz, Austria
[4] Department of Electronic and Electrical Engineering, University of Sheffield, Sheffield S1 3JD, UK
[*] Correspondence: danilo.riquelme.s@mail.pucv.cl

**Featured Application: The results of this work can be included in the design stage of modular permanent magnet synchronous machines for fault-tolerant applications.**

**Abstract:** Addressing stator-rotor misalignment, usually called eccentricity, is critical in permanent magnet (PM) machines since significantly high radial forces can be developed on the bearings, which can trigger a major fault and compromise the structural integrity of the machine. In this regard, this paper aims to provide insight into the unaddressed identification and analysis of the impact of eccentric tolerances on relevant performance indices of permanent magnet synchronous machines (PMSMs) with modular stator core. Static and dynamic eccentricity are assessed for different slot/pole combinations through the finite element method (FEM), and the results are compared with those of PMSMs with a conventional stator core. The unbalanced magnetic forces (UMF), cogging torque, back-emf, and mean torque variations are described and related to the eccentricity magnitude and classification. The main findings indicate that severe radial forces and significant additional cogging torque harmonics are generated because of eccentricity. Additionally, it is found that the main differences between modular PMSMs and conventional PMSMs rely on the value of slots per pole per phase.

**Keywords:** cogging torque; dynamic eccentricity; finite element analysis; modular stator; permanent magnet; static eccentricity; stator-rotor misalignment; tolerance analysis

## 1. Introduction

In safe-critical applications, the uninterrupted and reliable operation of the electrical machine is requisite, which is described by their fault-tolerance capability. This is the case, for instance, of aeronautical, aerospace, and electromobility applications [1–3]. As a concept, fault-tolerant machines should provide acceptable levels of electric, magnetic, physical, and thermal isolation of phase windings [3], and can also consider multiphase windings [4,5] or limiting short-circuit current capability [6,7].

In addition to the fault-tolerance capability, electrical machines must provide enough power density, efficiency, and electromagnetic performance to meet the application's requirements. In this regard, electrical machines based on permanent magnets (PMs) are well known as the most promising technology in terms of power density and efficiency [8–11], but they lack inherent fault-tolerance capability [12]. It is desirable that PMSMs resist winding open-circuit and short-circuit faults without compromising the integrity of magnets. Nevertheless, since their electromagnetic performance is unmatchable by other conventional topologies so far, considerable efforts have been carried out in the last few years to

enhance the fault-tolerant capability of PM machines [2–5,7,12–14]. In the technical literature, adopting a modular stator core, as illustrated in Figure 1b, has been shown to be a feasible solution to enhance the fault-tolerant capability of PM machines [15–23]. This obeys mechanical and electromagnetic reasons, since PM synchronous machines with modular stator core (MPMSMs) offer several advantages in high-power density and fault-tolerance applications [15]: they can provide physical, thermal, and electromagnetic insulation between phases, lower material requirements (laminations) than conventional topologies and a high slot fill factor [16]. Notwithstanding, the cost of MPMSMs developing these features is the presence of manufacturing and assembly challenges, decrease in mean torque, and increase in torque ripple [15,18,19]. As can be seen from Figure 1b, the stator structure in MPMSM is segmented and, therefore, the attachment of the modules to the stator body can provide the design with additional manufacturing and assemblies tolerances (uncertainties) when compared with conventional PMSMs (Figure 1a).

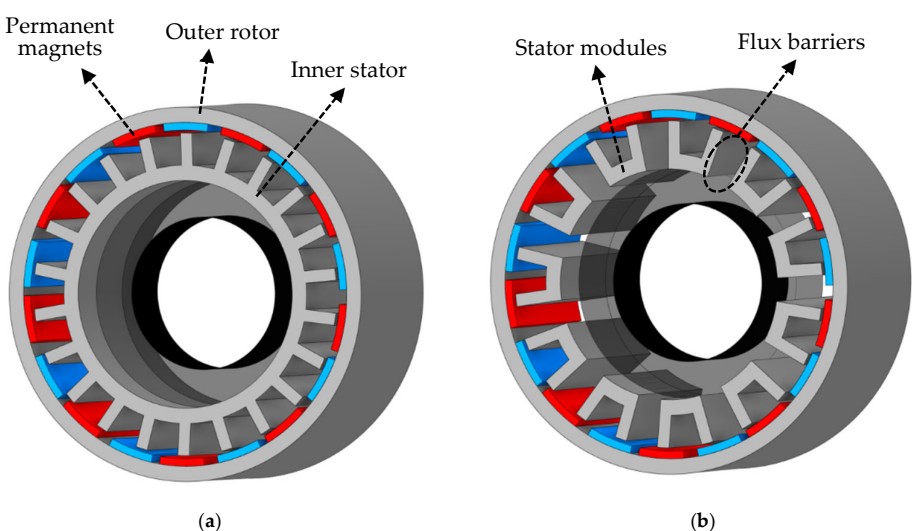

(**a**)  (**b**)

**Figure 1.** Three-dimensional representation of stator/rotor structure of a 24-slot, 20-pole PMSM: (**a**) conventional PMSM with non-segmented stator (monolithic), (**b**) PMSM with modular stator core (MPMSM).

Recent studies have approached the analysis of some of these uncertainties, giving insight into the effect of design parameters and dimensional tolerances on the machine performance [15,17,18]. Although these studies are relevant for describing the impact of dimensional tolerances on the performance of MPMSMs and provide guidelines for devising more robust designs, there is a critical uncovered aspect that compromises the reliability and continuous operation of these machines: stator-rotor misalignment.

Stator-rotor misalignment, which can be classified into static eccentricity or dynamic eccentricity depending on the dynamics of the misalignment [24,25], wears down the bearings since it generates vibration and pulsing forces on their structure [26,27] and can truncate the fault-tolerance capability of electrical machines. In addition, eccentricity affects the performance of several topologies in terms of the generation of radial forces in the rotor structure [27–29], cogging and ripple torque increase [24,30], input current harmonic distortion when the machine is fed with inverters [31], and back-emf unbalance [30].

Studies on eccentricity effects are particularly relevant in PM machines, as the magnets on the rotor can generate significantly high radial forces on the stator structure and the bearings [24], which can trigger a major fault. The impact of eccentricity on PMSMs with non-segmented stators has been covered in [25–27,29–31], focusing on quantifying and analyzing the effects on the electromagnetic performance of the machine. Particularly, [29] disclosed a crucial design tendency that can be included in the early stages of the machine design: the electromagnetic performance penalization due to eccentricity strongly depends

on the machine slot-pole combination, becoming especially relevant as the slot count and pole count get closer.

In turn, the impact of eccentricity on MPMSMs has been recently addressed in [24] for axial-even eccentricity in a single 24-slot, 28-pole machine: relevant radial forces appear as expected, but the cogging torque, back-emf, and electromagnetic torque seem to be insensitive. The authors concluded that the addressed slot/pole combination was mostly insensitive to eccentricity even when the slot count and pole count differed by four, but the study required to be expanded to more slot/pole combinations. The comparison between the results of [24,29] suggests that the effect of slot/pole combinations on PMSMs with eccentricity may be different from their modular counterpart. This could also be inferred from the analysis of [18], which states that, in presence of manufacturing tolerances, the periodicity of the stator core changes when adopting a modular structure, affecting the cogging torque main period and its sensitiveness to assembly tolerances for the same slot/pole combination.

All things considered, if the fault-tolerance capability of a MPMSM is to be assured, its electromagnetic performance and structure integrity should not be greatly affected by mild degrees of eccentricity. Nevertheless, and despite of its relevance, the effects of eccentricity on the performance of MPMSMs for different slot/pole combinations and their inclusion in the early design stages of the machine remains uncovered in the technical literature so far.

The aim of this work is, therefore, to identify and analyze the impact of eccentric tolerances on relevant performance indices of MPMSMs for different slot/pole combinations, comparing their response to that of PMSMs with conventional stators. Both static and dynamic eccentricity are analyzed by means of the finite element method (FEM) for five slot/pole combinations, evaluating the radial forces that the rotor structure is exposed to cogging torque, back-EMF, and rated torque. The relevance of this paper lies with providing a quick yet reliable comparison of the MPMSM performance between slot/pole combinations and the reasons behind the observed phenomena, critical in the design stage of a MPMSM, as well as disclosing relevant similarities and differences with respect to PMSMs with conventional stator core. This paper is organized as follows: in Section 2, the selected topology and its main data are presented; in Section 3, the methodology and details regarding the considered eccentricity types are described; in Section 4, the results of key performance indices when the machine has eccentricity tolerances are presented and discussed. The conclusions are drawn at the end of the paper.

## 2. Selected Topology and Slot/Pole Combinations

In order to compare the effect of eccentricity on different slot/pole combinations of both modular and non-modular PMSMs, tooth coil windings (TCW) were adopted since they allow a higher power/weight ratio due to the shorter end winding turns, lower cogging torque, and compactness when compared with the distributed windings [32]. The modular machine is considered to have U-Shaped stator segments and surface-mounted magnets as depicted in Figure 1b as they have been of interest in the last years for fault-tolerance applications [15,21]. The main data of the machines and their design parameters are presented in Table 1 and schematized in Figure 2. The machines assessed in this work aim to provide maximum mean torque within dimensional constraints, following the optimization criteria established in [17,21]. These machines represent low-power scaled protoypes with a rated power of 5 [kW], considering a current density of 10 A/mm$^2$ and a speed range of 0 to 6000 [RPM].

In turn, the slot/pole combinations were selected with the aim of covering different values of slots per phase per pole ($q$) given by:

$$q = \frac{Q_s}{2pm},$$

(1)

where $Q_s$ is the slot number, $2p$ is the pole number, and $m$ is the number of phases. For the case of TCW-PMSM, it is recommended to adopt a value of $q$ between 0.25 and 0.5 [32],

which is the reason why the selected slot/pole combinations are within this range. The combinations evaluated in this work are summarized in Table 2, which also shows their winding layouts. For all considered designs, the coil pitch is equal to 1 slot, as documented in [32].

**Table 1.** Main data of the machines (MPMSM and PMSM).

| Symbol | Quantity | Value |
|---|---|---|
| $d$ | Effective core length | 70.0 mm |
| $r_{se}$ | Stator outer radius | 133.0 mm |
| $\delta_g$ | Airgap length | 2.0 mm |
| $b_s$ | Slot width | 24.8 mm * |
| $h_s$ | Slot height | 27.5 mm |
| $b_t$ | Tooth width | 10.0 mm * |
| $h_{ys}$ | Height of the stator yoke | 10.5 mm |
| $h_{yr}$ | Height of the rotor yoke | 10.0 mm |
| $h_{pm}$ | PM height | 5.0 mm |
| $B_r$ | PM remanence | 1.1 T |
| $\mu_r$ | Relative recoil permeability | 1.04 |

* Reference tooth width and slot width values were considered for the 28-slot, 20-pole machines, as suggested in [17,18], and they were adjusted for another slot/pole combinations to develop similar saturation levels, based on [17,21].

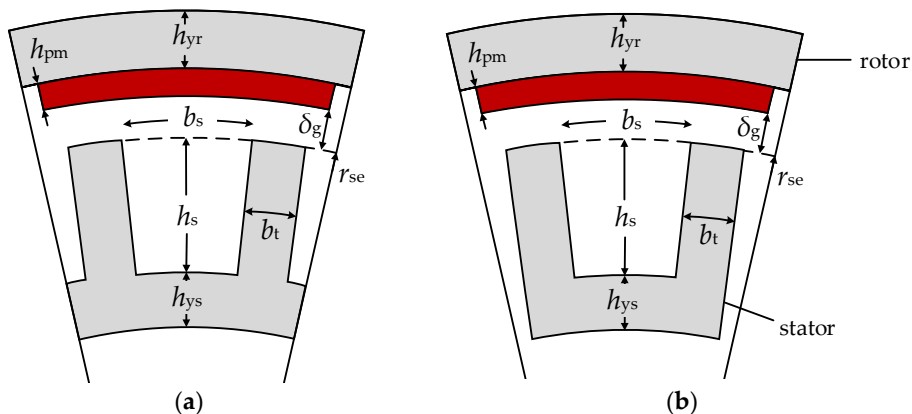

**Figure 2.** Schematics of selected PMSMs: (**a**) conventional PMSM with non-segmented stator, (**b**) PMSM with modular stator core.

**Table 2.** Selected slot/pole combinations, including their winding layout considering TCW.

| Slot Number | Pole Number | $q$ | $\Delta(Q_s, 2p)$ | Winding Layout |
|---|---|---|---|---|
| 18 | 12 | 1/2 | 6 | … \|A A′\|B B′\|C C′\| … |
| 18 | 20 | 3/10 | 2 | … \|C′ A\|A′ A′\|A A\|A′ B\|B′ B′\|B B\|B′ C\|C′ C′\|C C\| … |
| 24 | 20 | 2/5 | 4 | … \|C′ A\|A\|A′ A′\|A B′\|B B\|B′ C\|C′ C′\|C A′\|A A\|A′ B\| B′ B′\|B C′\|C C\| … |
| 24 | 22 | 4/11 | 2 | …A\|A′ A′\|A…A\|A′ A′\|A B′\|B B\|B′… B′\|B B\|B′ C\|C′ C′\|C … C\|C′ C′\|C… |
| 24 | 28 | 2/7 | 4 | … \|C′ A\|A′ A′\|A B′\|B B\|B′ C\|C′ C′\|C A′\|A A\|A′ B\| B′ B′\|B C′\|C C\| … |

In addition, machines with dissimilar differences between the slot number ($Q_s$) and the pole number ($2p$) are considered in this work from 2 to 6, as addressed in [33].

## 3. Types of Eccentricities Evaluated and Assessed Performance Indicators

In this research, both static eccentricity (SE) and dynamic eccentricity (DE) are assessed by means of FEM. When SE is present, and as depicted in Figure 3a, the rotor and stator structures are not coaxial and a non-uniform airgap distribution that does not vary with

the rotation of the rotor is generated. On the other hand, when DE is present, the rotor and stator are not coaxial and, additionally, the rotational axis and the rotor geometry axis do not match. As shown in Figure 3b, this creates a non-uniform airgap distribution that circulates with the rotor spinning.

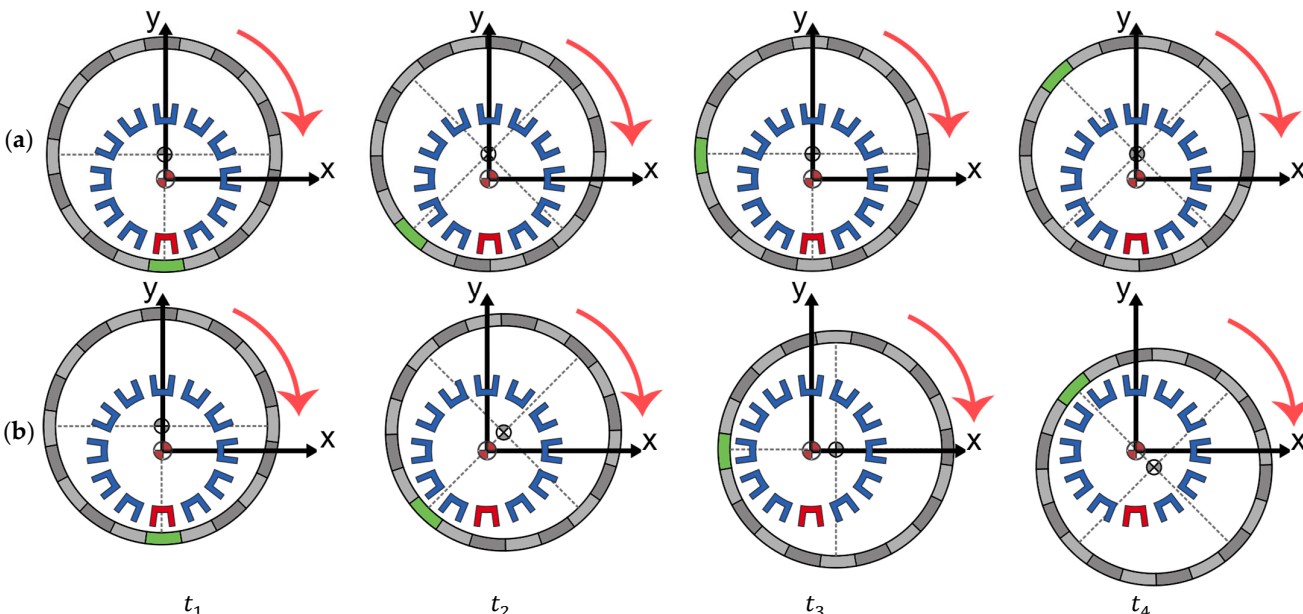

**Figure 3.** Representation of eccentricity types depending on its dynamics: (**a**) SE, in which the position of the minimum airgap is fixed, (**b**) DE, in which the position of the minimum airgap changes as the rotor structure rotates. Four arbitrary time instants are represented in the figure.

This work considers the difference between the nominal airgap length and the minimum airgap length generated by the stator/rotor misalignment as the magnitude of eccentricity as suggested in [24,30,33]. In addition, potential axis deflections and axial variations of the eccentricity magnitude are neglected. As a result, five eccentricity magnitudes are studied to cover from very low eccentricity (12.5% of the nominal airgap length) to high eccentricity (half the nominal airgap length), as presented in Table 3.

The effects of eccentricity on the performance of both PMSMs and MPMSMs are studied by means of: (i) the analysis of the generated radial forces, also called unbalanced magnetic force (UMF), (ii) cogging torque, (iii) back-emf, and (iv) mean torque, as these indicators have proven to be affected in conventional topologies for at least one slot/pole combination.

**Table 3.** Eccentricity magnitudes analyzed in the selected PMSMs.

| Eccentricity Magnitude (% of Nominal Airgap Length) | Eccentricity Magnitude (mm) | Severity |
|---|---|---|
| 12.5 | 0.25 | Very low |
| 25.0 | 0.50 | Low |
| 37.5 | 0.75 | Medium |
| 50.0 | 1.00 | High |

The following sections present the results organized so as to cover the impact of eccentricity on each performance indicator separately. In each case, the results are firstly presented in detail for one slot/pole combinations to show graphics of relevant curves and their harmonic spectrum. From these graphics, the main numerical indicators are extracted and summarized in tables for all the addressed slot/pole combinations, in order to allow

an organized comparison and analysis of results. All results are obtained by means of 2D FEM simulations carried out with the commercial package Ansys Electronics.

## 4. Results and Discussion: Unbalanced Magnetic Forces

One of the most critical consequences of eccentricity is the generation of unbalanced magnetic forces between the rotor and stator structures. In the case of radial-flux topologies, this may translate into radial forces of considerable magnitude that affect the bearings of the machine and whose dynamics depend on the eccentricity type. In this section, net forces acting on the rotor are assessed considering that in $t = 0$ the minimum airgap position is in $\theta = \frac{3}{2}\pi$ rad, as indicated in time instant $t_1$ of Figure 3.

### 4.1. Evaluation on a 24-Slot, 22-Pole PMSM and MPMSM

As an example, in Figure 4 the outcomes of the FEM evaluation of a 24-slot, 22-pole PMSM and a 24-slot, 22-pole MPMSM are presented for both SE and DE. The results were extracted considering the rotor structure completes one full turn. In Figure 4a,b the *x*-axis and *y*-axis forces acting on the rotor structure are shown when DE is present for the modular and the conventional machine, respectively. The X represents the average radial forces generated in a full rotation, which is zero for the case of DE. It may be noted that a rotating force vector is generated in both cases, with an almost constant magnitude and low ripple (visualized as saw teeth in the force circumference periphery).

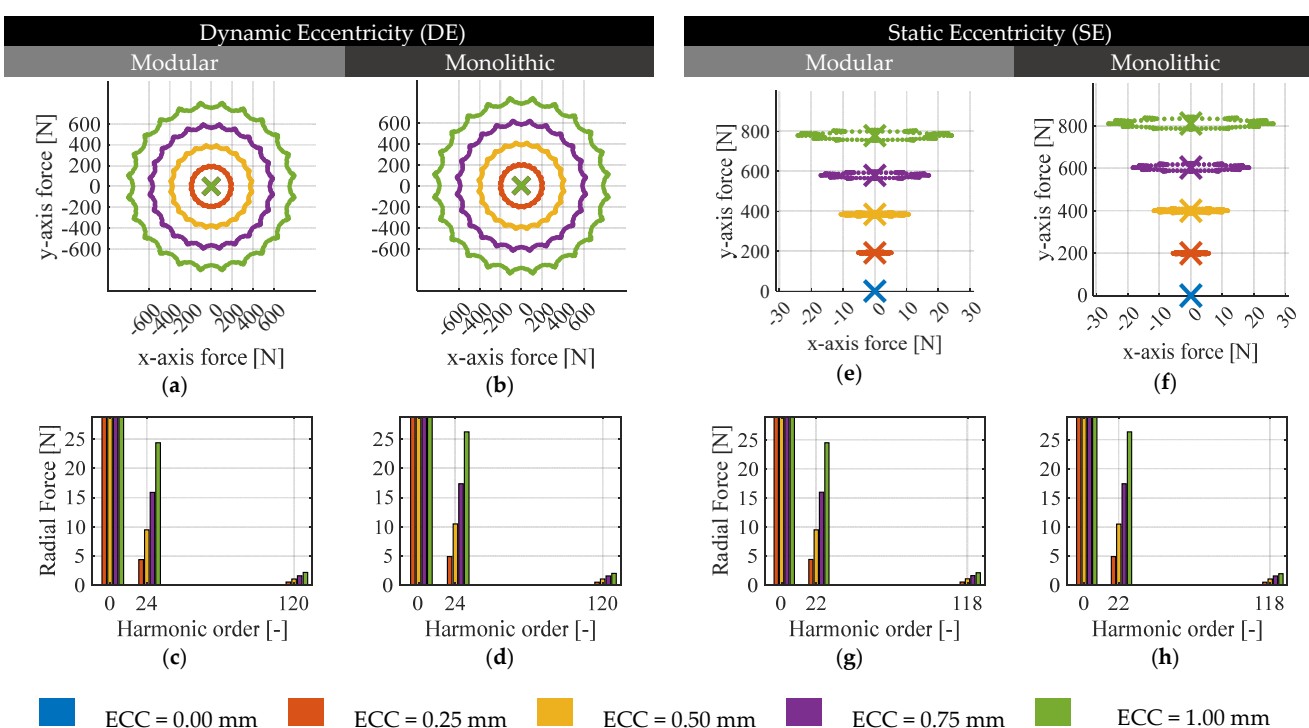

**Figure 4.** FE evaluation of radial forces acting on the rotor structure in the presence of DE (left column) and SE (right column) for a 24-slot, 22-pole PMSM and MPMSM: (**a**) *x*-axis and *y*-axis forces of DE on MPMSM, (**b**) *x*-axis and *y*-axis forces of DE on PMSM, (**c**) radial force spatial harmonic content of DE on MPMSM, (**d**) radial force spatial harmonic content of DE on PMSM, (**e**) *x*-axis and *y*-axis forces of SE on MPMSM, (**f**) *x*-axis and *y*-axis forces of SE on PMSM, (**g**) radial force spatial harmonic content of SE on MPMSM, (**h**) radial force spatial harmonic content of SE on PMSM.

From Figure 4c,d, which show the spatial harmonic content of the radial forces, it can be seen that the spatial frequency of the radial force ripple matches the slot pitch of the machine (HO = 24). This can be explained since for DE the relative position between

the magnet symmetry axis and teeth symmetry axis is periodical and repeats each $360/Q_s$ mechanical degrees. Similar findings were described for the cogging torque in conventional PMSMs in [34]. In turn, in Figure 4e,f, the *x*-axis and *y*-axis forces acting on the rotor structure are shown when SE is acting in the modular and the conventional machine, respectively. The X represents the average radial forces generated in a full rotation, which points from the minimum airgap position towards the center of rotation for the case of SE. It may be noted that the *x*-axis forces are not zero and configure the radial force ripple. From Figure 4g,h, which show the spatial harmonic content of the radial forces, it can be seen that the spatial frequency of the radial force ripple matches the pole pitch of the machine (HO = 22). This is because in SE the relative position between the magnet symmetry axis and teeth symmetry axis is periodical and repeats each $360/2p$ mechanical degrees, as described in [24].

In summary, relevant features of the radial force results can be summarized by the following indicators, useful for evaluating and comparing the different slot/pole combinations in Table 2.

- Mean value of the radial force acting on the rotor structure, measured in N.
- Peak-to-peak value of the radial force, representing the radial force ripple and measured in N.

### 4.2. Comparison of Slot/Pole Combinations for PMSMs and MPMSMs

In Table 4, the average forces generated by different DE magnitudes are presented for both the modular and monolithic machines considering different slot/pole combinations. In the first place, it may be noted that not all selected slot/pole combinations exhibit unbalanced magnetic forces in the absence of eccentricity. Secondly, and according to the tendencies of Figure 4a,b, it can be seen that the generated radial forces are directly proportional to the eccentricity magnitude. This means that the radial force created by eccentricity can be quickly predicted from a single ECC value.

**Table 4.** Average UMF of different PMSM and MPMSM slot/pole combinations in the presence of DE.

| ECC / Q/2p | 0 mm (Faultless) | | 0.25 mm | | 0.50 mm | | 0.75 mm | | 1.00 mm | |
|---|---|---|---|---|---|---|---|---|---|---|
| | Mod [N] | Mon [N] | Mod [N] | Mon [N] | Mod [N] | Mon [N] | Mod [N] | Mon [N] | Mod [N] | Mon [N] |
| 18S 12P | 0 | 0 | 92 | 116 | 184 | 231 | 277 | 348 | 370 | 466 |
| 18S 20P | 0 | 0 | 148 | 151 | 296 | 304 | 446 | 458 | 599 | 614 |
| 24S 20P | 0 | 0 | 181 | 194 | 362 | 390 | 546 | 588 | 733 | 790 |
| 24S 22P | 0 | 0 | 192 | 200 | 385 | 401 | 580 | 604 | 780 | 813 |
| 24S 28P | 0 | 0 | 203 | 209 | 408 | 419 | 617 | 634 | 833 | 856 |

From Table 4, it can be also noted that the unbalanced magnetic force increases as the slot number and pole number increase. The effect of the slot number can be noted from the comparison of the 18-slot, 20-pole machine to the 24-slot, 20-pole machine, which develops a consistent 20% force increase for all the evaluated ECC values. In consequence, it can be stated that the slot count has a medium impact on the radial forces generated by eccentricity and should not be neglected. In turn, the pole count effect can be deduced from the comparison of the 24-slot, 20-pole and the 24-slot, 28-pole machine, the latter having 12% more radial force. On the other hand, it is observed that in some slot/pole combinations, the radial force generated in the monolithic machine is significantly different from the modular machine. This can be explained by the flux density penalization of modular machines with a high value of *q* (See Equation (1)). To this end, Figure 5 shows the flux density lines for two slot/pole combinations that have different *q* values. Figure 5a,b show the flux lines when *q* is high, in which case the pole number is much smaller than the slot number. In this case, the flux lines require to travel through several teeth, crossing the stator yoke which is absent between the modules of MPMSMs. Therefore, in the

modular machine, the flux lines will cross the gap between modules, circulating across a high-reluctance path that penalizes the airgap flux density and, subsequently, the radial forces. Instead, Figure 5c,d show the flux lines when *q* is lower, in which case the pole number is closer to the slot number. In this situation, flux lines need to use fewer teeth to close the path, which is the reason why in modular machines the lines can circulate using a single module and not penalize the airgap flux density to a high extend.

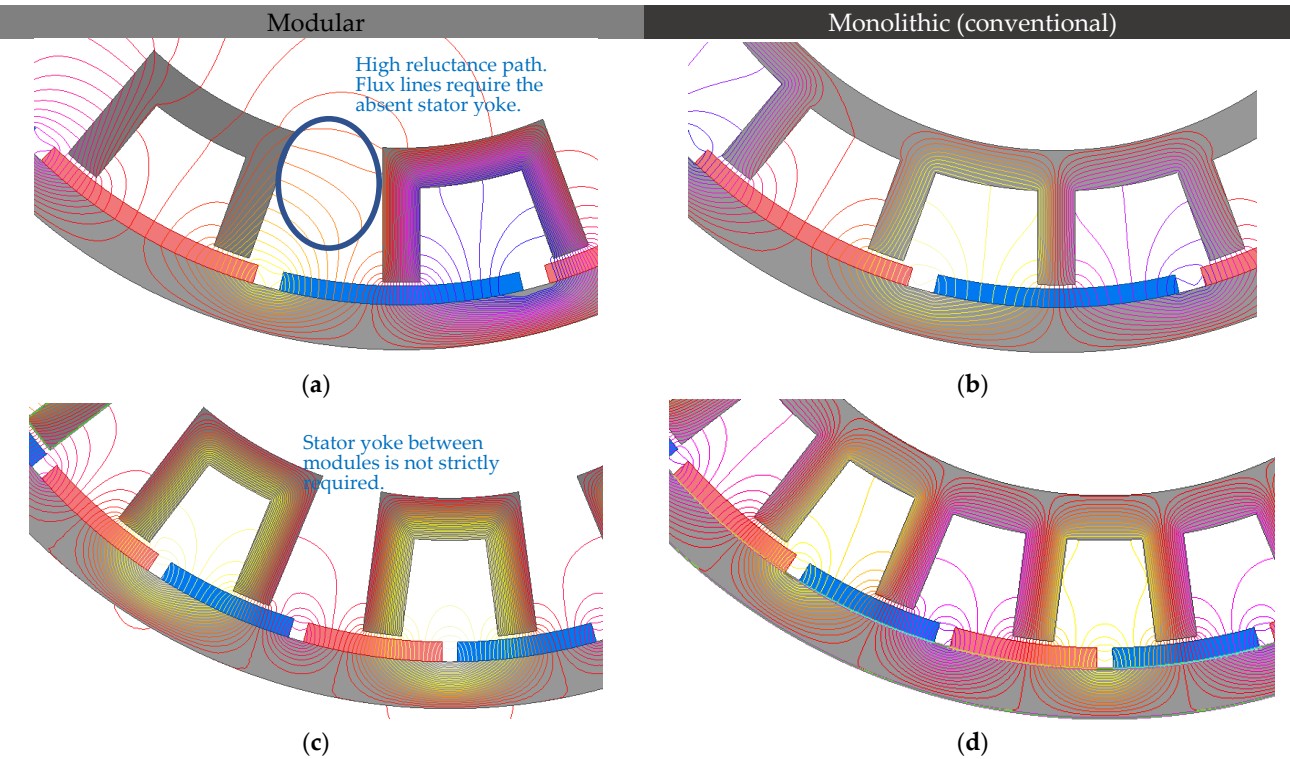

**Figure 5.** Close-up to flux density lines for different values of *q*: (**a**) Modular 18-slot, 12-pole, *q* = 0.5, (**b**) monolithic, 18-slot, 12-pole, *q* = 0.5, (**c**) modular, 24-slot, 22-pole, *q* = 0.36 (**d**) monolithic, 24-slot, 22-pole, *q* = 0.36.

In Table 5, the maximum radial force (ECC = 1 mm) for each slot/pole combination is presented and compared between modular and monolithic machines. It can be noted that, effectively, the difference between the radial forces generated on the modular machine vs. on the monolithic machine is higher as the value of *q* is greater.

**Table 5.** Difference between the radial force on modular and monolithic machines in terms of the number of slots per pole per phase.

| Q/2p | *q* | $F_{max}$ [N] **Mod** | $F_{max}$ [N] **Mon** | $\Delta F_{max}$ |
|------|-----|------|------|------|
| 18S 12P | 1/2 | 370 | 466 | 25.9% |
| 18S 20P | 3/10 | 599 | 614 | 2.5% |
| 24S 20P | 2/5 | 733 | 790 | 7.8% |
| 24S 22P | 4/11 | 780 | 813 | 4.2% |
| 24S 28P | 2/7 | 833 | 856 | 2.7% |

In Table 6, the radial force ripple generated by different magnitudes of DE is presented for each slot/pole combination. It is worth noting that slot/pole combinations with a lower difference between the slot count and the pole count exhibit the highest force ripple. This is particularly relevant since it translates into higher-frequency vibration and force pulses that can reach the bearings. For instance, for the 18-slot, 20-pole machines, the force

ripple is around 40% of the average radial force, which is already significant in magnitude, and it generates a force that pulses 18 times for each machine rotation. For all slot/pole combinations, the main harmonic order (HO) of the torque ripple matches the slot number, and there are no significant differences between the modular and the monolithic machines in this regard.

**Table 6.** Radial force ripple for different slot/pole combinations of PMSM and MPMSM in the presence of DE.

| ECC<br>Q/2p | 0 mm<br>(Faultless) | | 0.25 mm | | 0.50 mm | | 0.75 mm | | 1.00 mm | | Main HO | |
|---|---|---|---|---|---|---|---|---|---|---|---|---|
| | Mod<br>[N] | Mon<br>[N] | Mod<br>[N] | Mon<br>[N] | Mod<br>[N] | Mon<br>[N] | Mod<br>[N] | Mon<br>[N] | Mod<br>[N] | Mon<br>[N] | Mod<br>[-] | Mon<br>[-] |
| 18S 12P | 0 | 0 | 4 | 2 | 12 | 4 | 23 | 7 | 38 | 9 | 18 | 18 |
| 18S 20P | 0 | 0 | 81 | 83 | 160 | 164 | 235 | 242 | 304 | 314 | 18 | 18 |
| 24S 20P | 0 | 0 | 3 | 1 | 6 | 3 | 10 | 7 | 22 | 13 | 24 | 24 |
| 24S 22P | 0 | 0 | 10 | 11 | 22 | 24 | 37 | 39 | 55 | 58 | 24 | 24 |
| 24S 28P | 0 | 0 | 1 | 1 | 2 | 1 | 3 | 3 | 7 | 5 | 24 | 24 |

Table 7 summarizes the average value of the radial force generated by different SE magnitudes for both the modular and monolithic machines. In concordance with the tendencies of Figure 4e,f and similarly to DE, it can be seen that the generated radial forces are directly proportional to the eccentricity magnitude. Nevertheless, and as occurred in the case of DE, modular machines have significantly lower radial forces when the value of $q$ is high, for the reasons explained in Figure 5. From Table 7, it can also be seen that the unbalanced magnetic force increases as the slot number and pole number increase. Finally, when comparing Table 4 with Table 7, it may be appreciated that the force magnitude generated by DE is similar to that of SE.

**Table 7.** Average UMF for different slot/pole combinations of PMSM and MPMSM in the presence of SE.

| ECC<br>$Q_s/2p$ | $q$ | 0 mm (Faultless) | | 0.25 mm | | 0.50 mm | | 0.75 mm | | 1.00 mm | |
|---|---|---|---|---|---|---|---|---|---|---|---|
| | | Mod<br>[N] | Mon<br>[N] | Mod<br>[N] | Mon<br>[N] | Mod<br>[N] | Mon<br>[N] | Mod<br>[N] | Mon<br>[N] | Mod<br>[N] | Mon<br>[N] |
| 18S 12P | 1/2 | 0 | 0 | 92 | 115 | 183 | 231 | 276 | 348 | 370 | 466 |
| 18S 20P | 3/10 | 0 | 0 | 148 | 151 | 296 | 303 | 446 | 457 | 599 | 614 |
| 24S 20P | 2/5 | 0 | 0 | 180 | 194 | 362 | 389 | 546 | 587 | 733 | 790 |
| 24S 22P | 4/11 | 0 | 0 | 192 | 200 | 384 | 400 | 580 | 604 | 779 | 812 |
| 24S 28P | 2/7 | 0 | 0 | 203 | 208 | 408 | 419 | 617 | 634 | 833 | 855 |

In Table 8, the radial force ripple generated by different magnitudes of SE is presented for each slot/pole combination. Similar to DE, slot/pole combinations with a lower difference between the slot count and the pole count exhibit the highest force ripple. Again, the 18-slot, 20-pole machines provide the highest ripple of around 40% of the average radial forces. Different to what was observed for DE, in the case of SE, the main HO of the torque ripple matches the pole number, and there are no considerable differences between the modular and the monolithic machines.

**Table 8.** Radial force ripple for different slot/pole combinations of PMSM and MPMSM in the presence of SE.

| $Q_s/2p$ ECC | 0 mm (Faultless) | | 0.25 mm | | 0.50 mm | | 0.75 mm | | 1.00 mm | | Main HO | |
|---|---|---|---|---|---|---|---|---|---|---|---|---|
| | Mod [N] | Mon [N] | Mod [N] | Mon [N] | Mod [N] | Mon [N] | Mod [N] | Mon [N] | Mod [N] | Mon [N] | Mod [-] | Mon [-] |
| 18S 12P | 0 | 0 | 4 | 2 | 12 | 4 | 23 | 6 | 36 | 8 | 12 | 12 |
| 18S 20P | 0 | 0 | 81 | 83 | 160 | 165 | 236 | 244 | 307 | 318 | 20 | 20 |
| 24S 20P | 0 | 0 | 2 | 1 | 5 | 3 | 10 | 6 | 20 | 12 | 20 | 20 |
| 24S 22P | 0 | 0 | 11 | 12 | 22 | 24 | 37 | 40 | 55 | 60 | 22 | 22 |
| 24S 28P | 0 | 0 | 1 | 0 | 2 | 1 | 3 | 2 | 7 | 5 | 28 | 28 |

*4.3. Summary: DE and SE on the UMF of PMSMs and MPMSMs*

In summary, it was found that radial forces on machines with SE and DE scale with the slot number and pole number, and that the severity of these forces is lower in the case of modular machines depending on the value of *q*. Moreover, a force ripple of significant magnitude appears in machines in which the slot number and pole number are close to each other.

**5. Results and Discussion: Cogging Torque**

*5.1. DE: Evaluation of 24-Slot, 22-Pole PMSM and MPMSM*

In Figure 6, the cogging torque outcomes of the FE evaluation of a 24-slot, 22-pole PMSM and a 24-slot, 22-pole MPMSM are presented for DE. The results were extracted considering the rotor structure completes 360 mechanical degrees to correctly obtain the harmonic content of the cogging torque signal. However, only the main period (when eccentricity is present) is shown in the figure in order to provide a clearer visualization. In Figure 6a,c, the cogging torque waveforms are shown for a 24-slot, 22-pole MPMSM and PMSM respectively. In turn, Figure 6b,d present the harmonic spectrum of the cogging torque waveform for both machines, modular and conventional, when dynamic eccentricity (DE) is present.

The main period of the cogging torque was found to be 1.36 mechanical degrees when no eccentricity is applied in both cases. This changes when DE is present: the main period changes to 15 degrees, which corresponds to the slot pitch. This can be explained since in conventional PMSMs, the cogging torque main period is defined by the periodicities of the stator and rotor structures when no eccentricity is present (native harmonic content) and is accounted by [34]:

$$T_{T \text{ NHC}} = \frac{360}{\text{LCM}(Q_s, 2p)}. \tag{2}$$

This results in 1.36° for the 24-slot, 22-pole machine. However, when dimensional tolerances are affecting the machine, the stator/rotor periodicity breaks. For the case of DE, the minimum airgap is rotating with the rotor and the modules are stationary, and hence, the relative position between magnet symmetry axis and teeth symmetry axis is periodical and repeats each $360/Q_s$ mechanical degrees. Therefore, the main cogging torque period should change to the additional harmonic components generated, given by:

$$T_{T \text{ AHC DE}} = \frac{360}{Q_s}, \tag{3}$$

which happens to occur for the 24-slot, 22-pole machine and generates a significant increase in the peak-to-peak value of the cogging torque. The cogging torque after eccentricity

is therefore comprised of the native harmonic components (faultless) and the additional harmonic components (AHC) as per [34]:

$$T_{\text{cogg}}(\alpha) = T_{\text{NHC}}(\alpha) + T_{\text{AHC}}(\alpha) \tag{4}$$

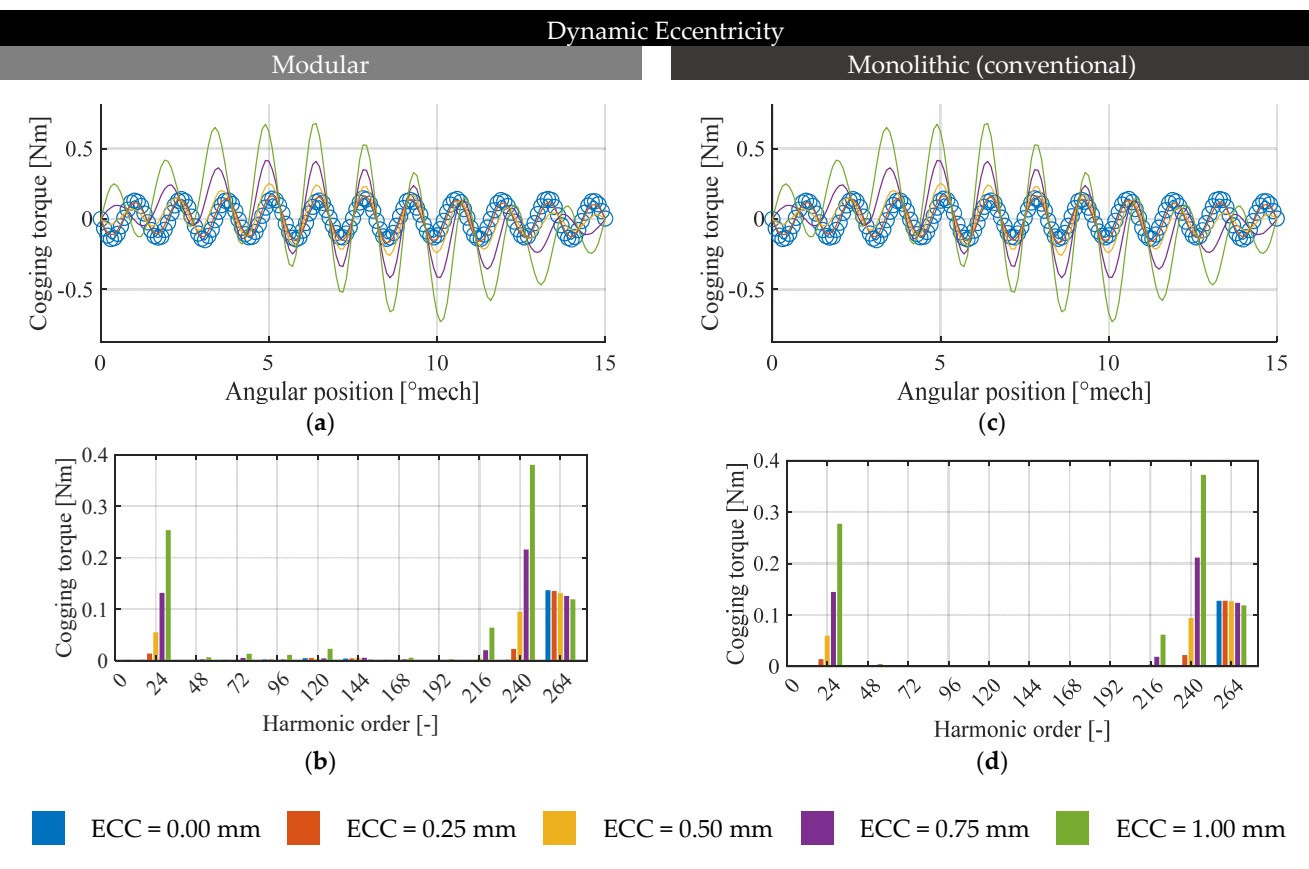

**Figure 6.** FE evaluation of cogging torque in the presence of DE for the 24-slot, 22-pole PMSM and MPMSM: (**a**) cogging torque waveform for the modular machine, (**b**) cogging torque spectrum for the modular machine, (**c**) cogging torque waveform for the monolithic machine, (**d**) cogging torque spectrum for the monolithic machine.

From Figure 6, it can be noted that significant cogging torque components with HO = 24 and their multiples are generated with DE, which results in a peak-to-peak value increase of up to 400%. Nevertheless, this does not hold true for all evaluated slot/pole combinations.

In summary, relevant features of the cogging torque results can be summarized by the following indicators, useful for evaluating and comparing the different slot/pole combinations of Table 2.

- NHC of the cogging torque, measured in Nm;
- AHC of the cogging torque, measured in Nm;
- AHC of the cogging torque, measured in percentage of the NHC.

*5.2. DE: Comparison of Slot/Pole Combinations for PMSMs and MPMSMs*

In Table 9, the natural harmonic component of cogging torque for different DE magnitudes is presented. Both the modular and monolithic machines are evaluated considering different slot/pole combinations. From Table 9, it may be noted that the cogging torque magnitude is very different depending on the slot/pole combination and it does not have a clear scalation with respect to the slot count or pole count separately. In this regard, it

is commonly accepted in faultless machines to consider the cogging torque HO (HO$_{NHC}$) as a comparative indicator for cogging torque magnitude ([35], see (2)). The higher the HO, the lower the main period of the cogging torque and the lower its magnitude should be. As can be seen from Table 9, the NHC of the cogging torque is barely affected by DE, and HO$_{NHC}$ is able to compare slot/pole combinations regardless of the eccentricity magnitude. It may draw the attention that a significant difference in the NHC magnitude can be observed for the 24-slot, 20-pole and the 24-slot, 28-pole machines when comparing the modular and the monolithic machine. This is related to a stator/rotor periodicity break generated by the modular structure of U-shape MPMSMs. In those cases, HO$_{NHC}$ should be corrected to include the periodicity provided by the modular stator core. As an example, in the case of the 24-slot, 28-pole machine, the main period of the cogging torque is 84 for the modular machine, which explains the difference regarding the monolithic machine.

**Table 9.** Cogging torque NHC of different PMSM and MPMSM-slot/pole combinations in the presence of DE.

| $Q_s/2p$ \ ECC | $HO_{NHC}$ | 0 mm (Faultless) | | 0.25 mm | | 0.50 mm | | 0.75 mm | | 1.00 mm | |
|---|---|---|---|---|---|---|---|---|---|---|---|
| | | Mod [Nm] | Mon [Nm] | Mod [Nm] | Mon [Nm] | Mod [Nm] | Mon [Nm] | Mod [Nm] | Mon [Nm] | Mod [Nm] | Mon [Nm] |
| 18S 12P | 72 | 19.11 | 22.42 | 19.13 | 22.44 | 19.17 | 22.46 | 19.24 | 22.49 | 19.34 | 22.54 |
| 18S 20P | 180 | 0.63 | 0.66 | 0.62 | 0.65 | 0.59 | 0.62 | 0.54 | 0.56 | 0.47 | 0.48 |
| 24S 20P | 120 | 3.92 | 2.70 | 3.91 | 2.71 | 3.88 | 2.77 | 3.84 | 2.86 | 3.83 | 2.99 |
| 24S 22P | 264 | 0.27 | 0.25 | 0.27 | 0.25 | 0.26 | 0.25 | 0.25 | 0.25 | 0.24 | 0.24 |
| 24S 28P | 168 | 2.28 | 0.72 | 2.28 | 0.73 | 2.30 | 0.74 | 2.34 | 0.77 | 2.40 | 0.81 |

In Table 10, the peak-to-peak value of the additional harmonic components of cogging torque for different DE magnitudes is summarized. Both the modular and monolithic machines are evaluated considering different slot/pole combinations. From Table 10, it is clear that the generated AHC of cogging torque is very different from a one slot/pole combination to another. Nevertheless, and contrary to what was observed for NHC, the AHC does not scale with HO$_{NHC}$. This can be explained by analyzing airgap flux density. As developed in [33], a simple analytical model of the additional airgap flux density radial and circumferential components accounting for the eccentricity can be expressed by:

$$B_{r_{ecc}} = B_{sr}\lambda_{ecc} \tag{5}$$

$$B_{\alpha\_ecc} = B_{s\alpha}\lambda_{ecc} \tag{6}$$

where $B_{sr}$ and $B_{s\alpha}$ are the radial and circumferential air flux densities (without the presence of eccentricity) with harmonic order $n$, and $\lambda_{ecc}$ is the real component of the equivalent complex permeance that represent the influence of eccentricity given by:

$$\lambda_{ecc} = 1 + \sum_v \lambda_v \cos(v\alpha - \chi v\omega t) \tag{7}$$

where $\lambda_v$ is the magnitude of the $v$-th order component of equivalent permeance representing the eccentricity and $\chi$ allows to select between SE ($\chi = 0$) and DE ($\chi = 1$). Since the stator slotting effect generates field harmonics of order $n = mp \pm \mu Q_s$, then the additional field harmonics due to eccentricity are $n \pm v$. By virtue of this, there are two scenarios in which eccentricity have an impact on the cogging torque:

- If both the additional flux density due to eccentricity and the flux density without eccentricity share harmonic spatial orders, eccentricity has an influence on the cogging torque;

- If the additional flux density due to eccentricity has a component of spatial order $l$ resulting from different values of $n$ and $v$, then eccentricity can also contribute to the cogging torque (for instance, if $l = n_1 \pm v_1 = n_2 \pm v_2$);
- In the case of machines having $2p = Q_s \pm 2$ (18-slot, 20-pole and 22-slot, 24-pole), the interaction between spatial harmonics from $n$ and $n \pm 2$ generate the AHC; in the case of machines that have $2p = Q_s \pm 4$ (24-slot, 20-pole and 28-slot, 24-pole), the interaction between spatial harmonics from $n$ and $n \pm 4$ generate the AHC; in the case of machines that have $2p = Q_s \pm 6$ (18-slot, 12-pole), the interaction between spatial harmonics from $n$ and $n \pm 6$ generate the AHC. It can be noted that several field harmonic components interact in machines with $2p = Q_s \pm 2$, and interacting harmonic components decrease as the difference between the slot number and the pole number is higher. The result of this analysis is that slot/pole combinations with the slot number being close to the pole number ($2p = Q_s \pm 2$) should have a more significant effect of eccentricity on cogging torque that machines with it slot number very different from its pole number ($2p = Q_s \pm 4$ and $2p = Q_s \pm 6$). This is verified from the results presented in Table 10.

**Table 10.** Peak-to-peak value of cogging torque AHC of different PMSM and MPMSM slot/pole combinations in the presence of DE.

| $Q_s$/$2p$  ECC | $HO_{NHC}$ | $\Delta(Q_s, 2p)$ | 0.25 mm | | 0.50 mm | | 0.75 mm | | 1.00 mm | |
|---|---|---|---|---|---|---|---|---|---|---|
| | | | Mod [Nm] | Mon [Nm] | Mod [Nm] | Mon [Nm] | Mod [Nm] | Mon [Nm] | Mod [Nm] | Mon [Nm] |
| 18S 12P | 72 | 6 | 0.06 | 0.06 | 0.12 | 0.15 | 0.26 | 0.29 | 0.45 | 0.48 |
| 18S 20P | 180 | 2 | 0.20 | 0.20 | 0.74 | 0.78 | 1.59 | 1.70 | 2.80 | 2.95 |
| 24S 20P | 120 | 4 | 0.04 | 0.06 | 0.13 | 0.09 | 0.32 | 0.22 | 0.63 | 0.50 |
| 24S 22P | 264 | 2 | 0.08 | 0.09 | 0.30 | 0.31 | 0.72 | 0.73 | 1.39 | 1.36 |
| 24S 28P | 168 | 4 | 0.02 | 0.02 | 0.05 | 0.04 | 0.13 | 0.09 | 0.29 | 0.21 |

The results of the 18-slot, 12-pole machines versus that of the 24-slot, 28-pole machines may draw attention since they do not exactly follow this tendency: the 18-slot, 12-pole machine has higher absolute AHC than the 24-slot, 28-pole machine, but it has a higher difference between the slot count and the pole count. However, the analysis must consider the proportion between the AHC generated by eccentricity and the NHC. From Table 9, it can be recalled that the 18-slot, 12-pole machine has a significantly higher cogging torque NHC than the 24-slot, 28-pole machine, which translates into AHC being proportionally lower than that of the 24-slot, 28-pole machine.

Table 11 is created to account for this proportion and presents the relative cogging torque increase of the different slot/pole combinations in the presence of DE. It may be appreciated that the relative cogging torque increase effectively depends on the difference between the slot count and the pole count, as indicated in the spatial field harmonic analysis. Cogging torque of modular and monolithic machines have a similar response to eccentricity, although some slot/pole combinations have mild differences due to the break of stator/rotor periodicity, as indicated in [18].

**Table 11.** Relative cogging torque increase of different PMSM and MPMSM slot/pole combinations in the presence of DE.

| $Q_s/2p$ \ ECC | $\Delta(Q_s, 2p)$ | 0.25 mm | | 0.50 mm | | 0.75 mm | | 1.00 mm | |
|---|---|---|---|---|---|---|---|---|---|
| | | Mod [%] | Mon [%] | Mod [%] | Mon [%] | Mod [%] | Mon [%] | Mod [%] | Mon [%] |
| 18S 12P | 6 | 0.19 | 0.22 | 0.53 | 0.65 | 1.15 | 1.30 | 2.07 | 2.18 |
| 18S 20P | 2 | 27.23 | 26.21 | 102.72 | 105.80 | 238.69 | 236.47 | 422.04 | 424.21 |
| 24S 20P | 4 | 0.29 | 0.69 | 1.33 | 3.05 | 4.42 | 7.62 | 11.00 | 17.45 |
| 24S 22P | 2 | 12.58 | 16.57 | 72.73 | 89.18 | 180.61 | 215.49 | 374.00 | 435.19 |
| 24S 28P | 4 | 0.41 | 0.11 | 1.62 | 2.12 | 4.64 | 8.61 | 10.33 | 26.39 |

*5.3. SE: Evaluation on 24-Slot, 22-Pole PMSM and MPMSM*

In Figure 7, the cogging torque outcomes of the FE evaluation of a 24-slot, 22-pole PMSM and a 24-slot, 22-pole MPMSM are presented for SE. Similar to DE results, SE results were extracted considering the rotor structure completes one full turn to correctly obtain the harmonic content of the cogging torque signal. However, only the main period (when eccentricity is present) is presented in order to provide a clearer visualization. In Figure 7a,c, the cogging torque waveform is shown for a 24-slot, 22-pole MPMSM and PMSM, respectively. In turn, Figure 7b,d present the harmonic spectrum of the cogging torque waveform for both machines, modular and conventional, when SE is present.

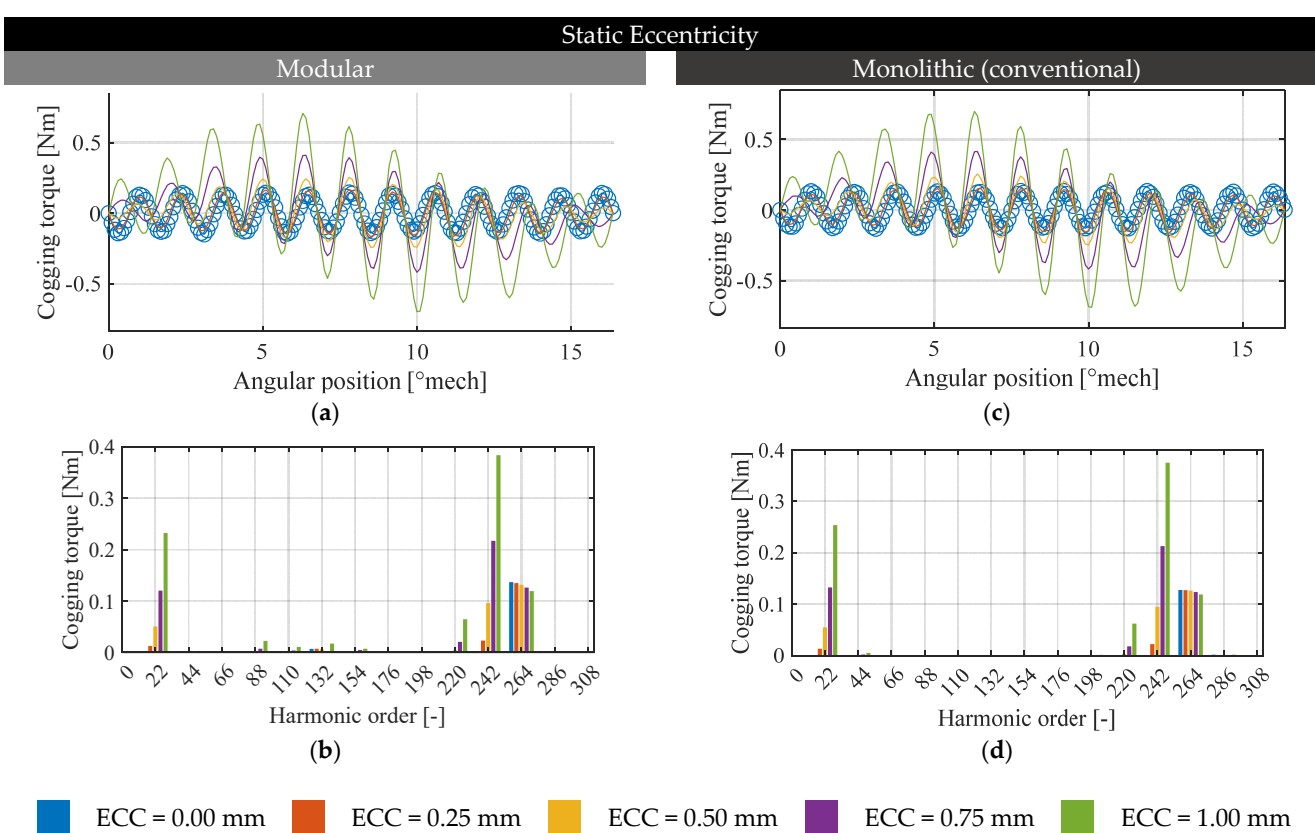

**Figure 7.** FE evaluation of cogging torque in the presence of SE for 24-slot, 22-pole PMSM and MPMSM: (**a**) cogging torque waveform for the modular machine, (**b**) cogging torque spectrum for the modular machine, (**c**) cogging torque waveform for the monolithic machine, (**d**) cogging torque spectrum for the monolithic machine.

When SE is present, the main period of the cogging torque changes from 1.36 to 16.36 mechanical degrees, which corresponds to the pole pitch. From (2), it may be noted that the cogging torque main period is defined by the periodicities of the stator and rotor structures when no eccentricity is present, but eccentricity breaks the stator/rotor periodicity. For the case of SE, the minimum airgap is stationary, and the relative position between the magnet symmetry axis and teeth symmetry axis depends exclusively on the movement of the poles, which is periodical and repeats each $360/2p$ mechanical degrees. Therefore, the main cogging torque period should change to the additional harmonic components generated, given by:

$$T_{T \text{ AHC SE}} = \frac{360}{2p}, \tag{8}$$

From Figure 7, it can be noted that significant cogging torque components with HO = 22 and its multiples are generated with SE, which results in a peak-to-peak value increase of up to 400%. Nevertheless, this does not hold true for all evaluated slot/pole combinations.

### 5.4. SE: Comparison of Slot/Pole Combinations for PMSMs and MPMSMs

In Table 12, the natural harmonic component of cogging torque for different SE magnitudes is presented. Both the modular and monolithic machines are evaluated considering different slot/pole combinations. From Table 12, it can be seen that the NHC of the cogging torque is barely affected by SE and, therefore, the natural cogging torque HO ($T_{T \text{ NHC}}$) can be used as a comparative indicator for cogging torque magnitude. Similar to the results obtained for SE, from Table 12, it can be noted that $HO_{NHC}$ is able to compare slot/pole combinations regardless of the eccentricity magnitude.

**Table 12.** Cogging torque NHC of different PMSM and MPMSM slot/pole combinations in the presence of SE.

| $Q_s/2p$ ＼ ECC | $HO_{NHC}$ | 0 mm (Faultless) | | 0.25 mm | | 0.50 mm | | 0.75 mm | | 1.00 mm | |
|---|---|---|---|---|---|---|---|---|---|---|---|
| | | Mod [Nm] | Mon [Nm] | Mod [Nm] | Mon [Nm] | Mod [Nm] | Mon [Nm] | Mod [Nm] | Mon [Nm] | Mod [Nm] | Mon [Nm] |
| 18S 12P | 72 | 19.10 | 22.42 | 19.13 | 22.44 | 19.16 | 22.46 | 19.20 | 22.49 | 19.24 | 22.54 |
| 18S 20P | 180 | 0.63 | 0.66 | 0.62 | 0.66 | 0.60 | 0.63 | 0.57 | 0.59 | 0.52 | 0.54 |
| 24S 20P | 120 | 3.90 | 2.66 | 3.90 | 2.68 | 3.87 | 2.74 | 3.83 | 2.82 | 3.82 | 2.96 |
| 24S 22P | 264 | 0.27 | 0.25 | 0.27 | 0.25 | 0.26 | 0.25 | 0.25 | 0.25 | 0.24 | 0.24 |
| 24S 28P | 168 | 2.28 | 0.72 | 2.28 | 0.73 | 2.30 | 0.75 | 2.34 | 0.78 | 2.40 | 0.81 |

In Table 13, the relative cogging increase of the cogging torque for different SE magnitudes is summarized, which is calculated as the ratio between the peak-to-peak value of the AHC and the NHC. Both the modular and monolithic machines are evaluated considering different slot/pole combinations. From Table 13, it is clear that the generated AHC of cogging torque is very different from a slot/pole combination to another. It may be appreciated that the relative cogging torque increase depends on $\Delta(Q_s, 2p)$ in a similar manner to what was observed for DE (see Table 11). This tendency is defined by the spatial field harmonics that interact to generate cogging torque for each machine as per (5) to (7). Furthermore, the cogging torque of modular and monolithic machines have a similar response to eccentricity.

**Table 13.** Relative cogging torque increase of different PMSM and MPMSM slot/pole combinations in the presence of SE.

| $Q_s/2p$    ECC | $HO_{NHC}$ | $\Delta(Q_s, 2p)$ | 0.25 mm | | 0.50 mm | | 0.75 mm | | 1.00 mm | |
|---|---|---|---|---|---|---|---|---|---|---|
| | | | Mod [%] | Mon [%] | Mod [%] | Mon [%] | Mod [%] | Mon [%] | Mod [%] | Mon [%] |
| 18S 12P | 72 | 6 | 0.19 | 0.22 | 0.54 | 0.66 | 1.17 | 1.31 | 2.05 | 2.19 |
| 18S 20P | 180 | 2 | 28.92 | 24.87 | 117.38 | 107.98 | 260.77 | 244.69 | 462.91 | 439.94 |
| 24S 20P | 120 | 4 | 0.41 | 0.83 | 1.52 | 2.89 | 4.63 | 7.72 | 11.22 | 17.86 |
| 24S 22P | 264 | 2 | 12.31 | 16.10 | 67.51 | 87.25 | 178.98 | 210.45 | 373.15 | 417.45 |
| 24S 28P | 168 | 4 | 0.55 | 0.59 | 1.46 | 2.72 | 4.37 | 9.48 | 10.14 | 25.80 |

*5.5. Summary: SE and DE on Cogging Torque of PMSMs and MPMSMs*

In summary, it was found that the cogging torque can be separated into two components: NHC, which are originally present in the faultless machine, and AHC, which appears due to eccentricity. In this regard:

- NHC is not affected by eccentricity, and its magnitude can be related to the slot/pole combination by means of the main harmonic order of the cogging torque, which can be determined as the least common multiple of slot and pole number.
- AHC is affected by eccentricity and can translate into severe peak-to-peak increases of the cogging torque. Machines that have a slot number close to their pole number are very sensitive to eccentricity and develop a higher cogging torque increase. In terms of magnitude, there are no significant differences between DE and SE. However, the order of the AHC generated by DE are multiples of the slot number, and the order of the AHC generated by SE are multiples of the pole number.
- Modular machines may have different native harmonic components when compared to their monolithic counterparts due to the stator/rotor periodicity break when adopting a modular stator core. Nevertheless, AHC is generated to a similar extent in both modular and monolithic machines.

These findings can be used for diagnosis ends: in machines with their slot number close to their pole number, anomalous cogging torque increases with certain harmonic orders can indicate the presence of SE or DE.

**6. Results and Discussion: Back-Emf**

*6.1. DE: Evaluation on a 24-Slot, 22-Pole PMSM and MPMSM*

In Figure 8, the back-emf outcomes of the FE evaluation of a 24-slot, 22-pole PMSM and a 24-slot, 22-pole MPMSM are presented for DE. The results were extracted considering the rotor structure completes one full turn to correctly obtain the harmonic content of the back-emf signal, which is depicted in the figures. In Figure 8a,c, the three-phase back-emf waveforms are presented for a 24-slot, 22-pole MPMSM and PMSM, respectively. In turn, Figure 8b,d show the harmonic spectrum of the back-emf for both machines, modular and conventional, when DE is present.

From Figure 8, it can be noted that the back-emf waveform is composed of a high magnitude fundamental component and a third-order harmonic component of low magnitude. In addition, it is evident that eccentricity does not have a significant impact on the back-emf magnitude, and no significant unbalance between phases is observed either, contrary to what was suggested in [30].

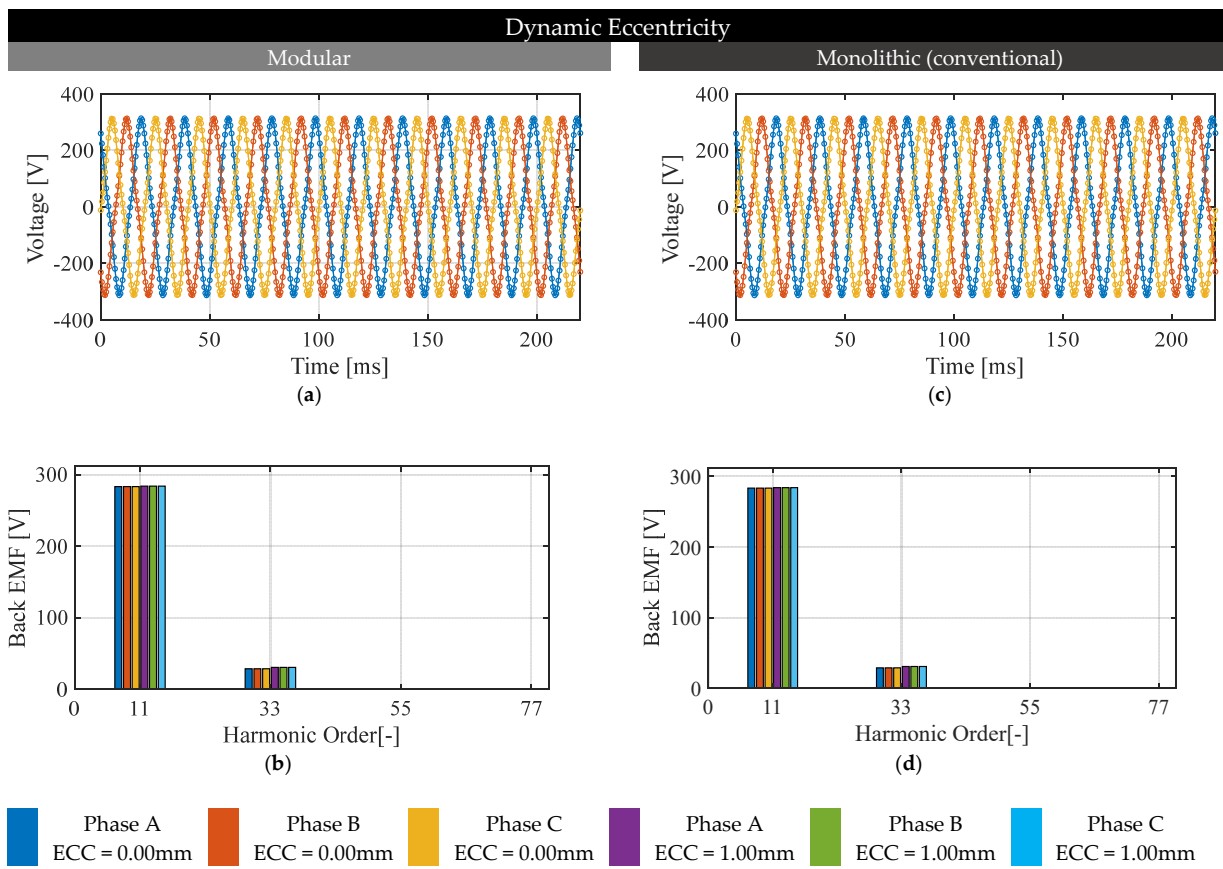

**Figure 8.** FE evaluation of the back-emf in the presence of DE for a 24-slot, 22-pole PMSM and MPMSM: (**a**) back-emf waveform for the modular machine, (**b**) back-emf spectrum for the modular machine, (**c**) back-emf waveform for the monolithic machine, (**d**) back-emf spectrum for the monolithic machine.

A significant back-emf unbalance was observed in [30] when evaluating eccentricity on a 9-slot, 8-pole PMSM with conventional stator core. This can be explained by analyzing the spatial distribution of phases of that 9-slot, 8-pole machine when compared to an 8-slot, 12-pole PMSM, which are represented in Figure 9a,b, respectively.

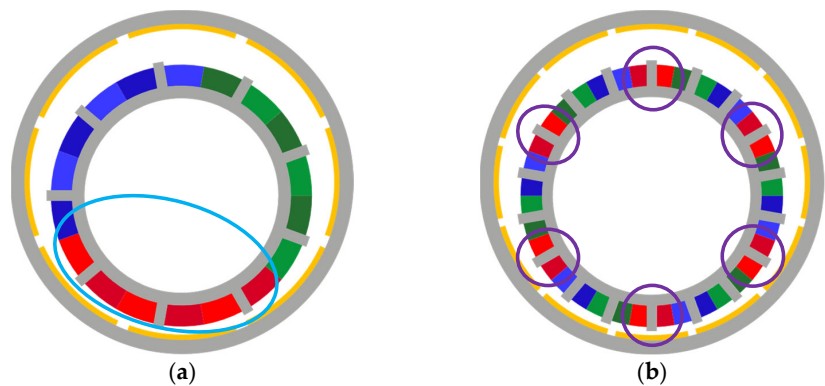

**Figure 9.** Schematics of (**a**) a 9-slot, 8-pole PMSM with conventional stator core subject to eccentricity. In this case, the phase windings form a single group of conductors (light blue oval) distributed in a specific part of the stator circumference and (**b**) an 18-slot, 12-pole PMSM with conventional stator core subject to eccentricity. In this case, the phase windings form six group of conductors (light blue oval) evenly distributed in the stator circumference, damping the potential back-emf unbalance.

For the case of the 9-slot, 8-pole PMSM, when eccentricity is applied, the minimum airgap position can increase or decrease the back-emf magnitude of a phase depending on how close it is to the phase windings, since it can strengthen or weaken the linked flux. As can be seen from the schematics of Figure 9a, phase A is strengthened as the airgap in front of phase A is smaller, and the back-emf of phase B and C should be lower since the equivalent airgap is larger than the original. In that case, the phase windings are not evenly distributed in the stator circumference, which translates into an unbalanced back-emf waveform when eccentricity is present. By the contrary, in the case of the 18-slot, 12-pole machine shown in Figure 9b, the phase windings are forming six groups of conductors evenly distributed in the stator circumference. Even when the rotor and stator are misaligned, the flux linked by a phase is strengthened on one side of the machine but weakened on the other side of the machine, which results into a low-to-null variation of the back-emf. The number of evenly distributed phase groups will be denoted by $N_{pg}$ from now on.

In summary, relevant features of the back-emf results can be summarized by the following indicators, useful to evaluate and compare the different slot/pole combinations of Table 2.

- Fundamental component of the back-emf, measured in V;
- Third-order harmonic of the back-emf, measured in V;
- Maximum unbalance between phases, measured in V.

### 6.2. DE: Comparison of Slot/Pole Combinations for PMSMs and MPMSMs

In Table 14, the fundamental component of back-emf in the presence of the maximum DE magnitude (1.00 mm) is presented for both the modular and monolithic machines considering different slot/pole combinations. As in the case of the 24-slot, 22-pole machine, no significant effect is observed in the back-emf fundamental and third-order harmonics caused by eccentricity.

**Table 14.** Fundamental component and third-order harmonic of back-emf for different PMSM and MPMSM slot/pole combinations in the presence of DE.

| ECC $Q_s/2p$ | | 0 mm (Faultless) | | | | | | 1.00 mm | | | | | |
|---|---|---|---|---|---|---|---|---|---|---|---|---|---|
| | | Modular [V] | | | Monolithic [V] | | | Modular [V] | | | Monolithic [V] | | |
| | HO | A | B | C | A | B | C | A | B | C | A | B | C |
| 18S 12P | 1st | 247.5 | 247.5 | 247.5 | 263.7 | 263.8 | 263.7 | 247.6 | 247.7 | 247.6 | 263.7 | 263.9 | 263.7 |
| | 3rd | 21.9 | 21.9 | 21.9 | 24.6 | 24.0 | 24.6 | 22.0 | 22.0 | 22.0 | 24.8 | 24.4 | 24.9 |
| 18S 20P | 1st | 281.0 | 281.1 | 281.0 | 275.4 | 275.5 | 275.4 | 281.7 | 281.7 | 281.6 | 276.2 | 276.2 | 276.2 |
| | 3rd | 31.6 | 31.6 | 31.6 | 35.9 | 36.0 | 36.0 | 33.3 | 33.3 | 33.3 | 37.8 | 37.8 | 37.8 |
| 24S 20P | 1st | 276.7 | 276.7 | 276.6 | 275.9 | 275.9 | 275.8 | 277.5 | 277.5 | 277.5 | 276.7 | 276.6 | 276.6 |
| | 3rd | 37.9 | 37.9 | 37.9 | 37.8 | 37.8 | 37.9 | 40.5 | 40.6 | 40.6 | 39.7 | 39.7 | 39.7 |
| 24S 22P | 1st | 283.4 | 283.4 | 283.3 | 283.2 | 283.2 | 283.2 | 284.1 | 284.1 | 284.1 | 283.9 | 283.9 | 283.9 |
| | 3rd | 28.7 | 28.7 | 28.7 | 29.0 | 28.9 | 29.0 | 30.7 | 30.7 | 30.7 | 30.8 | 30.8 | 30.9 |
| 24S 28P | 1st | 300.9 | 300.9 | 300.9 | 292.5 | 292.5 | 292.5 | 303.4 | 303.4 | 303.4 | 294.8 | 294.8 | 294.8 |
| | 3rd | 9.5 | 9.5 | 9.7 | 18.9 | 18.9 | 19.1 | 9.6 | 9.6 | 9.8 | 19.8 | 19.8 | 19.8 |

In Table 15, the maximum unbalance between phases in the presence of the maximum DE magnitude (1.00 mm) is presented for both the modular and monolithic machines considering different slot/pole combinations. It can be noted that the number of phase groups evenly distributed in the stator circumference is a strong indicator of the magnitude of back-emf unbalance for a given slot/pole combination. No significant differences were found by comparing the results of modular and monolithic machines.

**Table 15.** Maximum back-emf unbalance for different PMSM and MPMSM slot/pole combinations in the presence of DE.

| ECC | Modular [V] | Monolithic [V] | $N_{pg}$ |
|---|---|---|---|
| 18S 12P | 0.526 | 0.657 | 6 |
| 18S 20P | 10.386 | 9.091 | 2 |
| 24S 20P | 1.331 | 1.312 | 4 |
| 24S 22P | 6.660 | 6.286 | 2 |
| 24S 28P | 1.065 | 1.301 | 4 |

*6.3. SE: Evaluation on a 24-Slot, 22-Pole PMSM and MPMSM*

In Figure 10, the back-emf outcomes of the FE evaluation of a 24-slot, 22-pole PMSM and a 24-slot, 22-pole MPMSM are presented for SE. The results were extracted considering the rotor structure completes one full turn to correctly obtain the harmonic content of the back-emf signal, which is depicted in the figures. In Figure 10a,c, the three-phase back-emf waveforms are presented for a 24-slot, 22-pole MPMSM and PMSM, respectively. In turn, Figure 10b,d show the harmonic spectrum of the back-emf for both machines, modular and conventional, when SE is present.

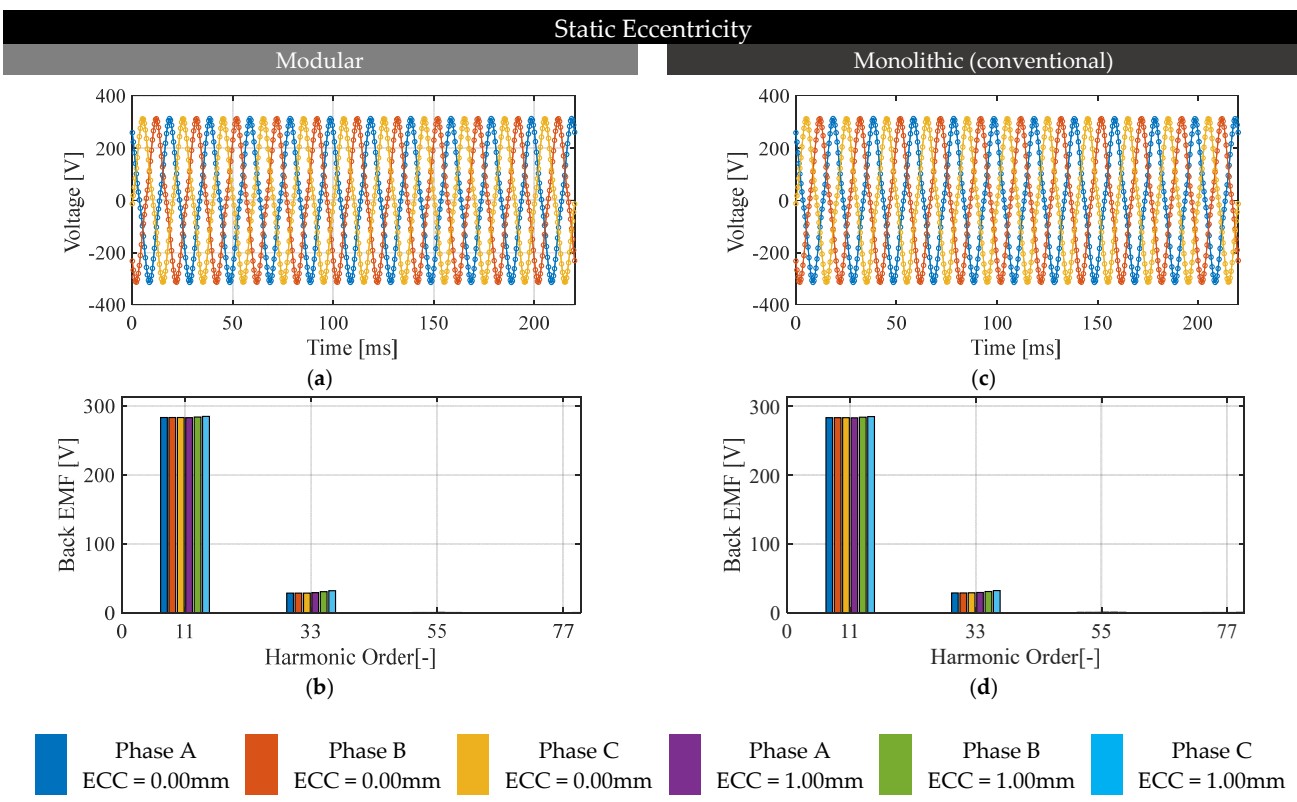

**Figure 10.** FE evaluation of the back-emf in the presence of SE for a 24-slot, 22-pole PMSM and MPMSM: (**a**) back-emf waveform for the modular machine, (**b**) back-emf spectrum for the modular machine, (**c**) back-emf waveform for the monolithic machine, (**d**) back-emf spectrum for the monolithic machine.

From Figure 10, and similar to what was observed in Figure 8, eccentricity does not have a significant impact on the back-emf magnitude, and no significant unbalance between phases is observed either.

### 6.4. SE: Comparison of Slot/Pole Combinations for PMSMs and MPMSMs

In Table 16, the fundamental component of back-emf in the presence of the maximum SE magnitude (1.00 mm) is presented for both the modular and monolithic machines considering different slot/pole combinations. As in the case of the 24-slot, 22-pole machine, no significant effect is observed in the back-emf fundamental and third-order harmonics caused by eccentricity.

**Table 16.** Fundamental component and third-order harmonic of back-emf for different PMSM and MPMSM slot/pole combinations in the presence of SE.

| ECC $Q_s/2p$ | | 0 mm (Faultless) | | | | | | 1.00 mm | | | | | |
|---|---|---|---|---|---|---|---|---|---|---|---|---|---|
| | | Modular [V] | | | Monolithic [V] | | | Modular [V] | | | Monolithic [V] | | |
| | HO | A | B | C | A | B | C | A | B | C | A | B | C |
| 18S 12P | 1st | 247.2 | 247.3 | 247.2 | 263.5 | 263.7 | 263.5 | 247.5 | 247.6 | 247.4 | 263.7 | 263.8 | 263.7 |
| | 3rd | 21.9 | 21.9 | 21.9 | 24.5 | 24.1 | 24.5 | 21.9 | 22.0 | 22.1 | 24.6 | 24.6 | 24.6 |
| 18S 20P | 1st | 280.7 | 280.8 | 280.7 | 275.1 | 275.2 | 275.1 | 281.2 | 280.6 | 283.0 | 275.3 | 275.4 | 277.5 |
| | 3rd | 31.7 | 31.7 | 31.7 | 36.0 | 36.1 | 36.1 | 30.5 | 32.9 | 36.97 | 36.2 | 36.2 | 41.3 |
| 24S 20P | 1st | 276.7 | 276.8 | 276.7 | 281.7 | 281.7 | 281.7 | 277.4 | 277.4 | 277.3 | 282.3 | 282.3 | 282.2 |
| | 3rd | 37.9 | 37.8 | 37.9 | 38.6 | 38.6 | 38.6 | 40.3 | 40.4 | 40.4 | 40.4 | 40.4 | 40.5 |
| 24S 22P | 1st | 283.4 | 283.4 | 283.3 | 283.2 | 283.2 | 283.2 | 283.1 | 284.0 | 285.0 | 282.9 | 283.9 | 284.9 |
| | 3rd | 28.7 | 28.7 | 28.7 | 29.0 | 29.0 | 29.0 | 29.3 | 30.8 | 32.2 | 29.6 | 31.0 | 32.2 |
| 24S 28P | 1st | 300.9 | 301.0 | 300.9 | 292.5 | 292.5 | 292.5 | 303.2 | 303.2 | 303.1 | 294.6 | 294.6 | 294.5 |
| | 3rd | 9.5 | 9.5 | 9.6 | 19.0 | 19.0 | 19.1 | 9.7 | 9.7 | 9.7 | 19.8 | 19.8 | 19.8 |

In Table 17, the maximum unbalance between phases in the presence of the maximum SE magnitude (1.00 mm) is presented for both the modular and monolithic machines considering different slot/pole combinations. As in the case of DE, it is clear that the number of phase groups evenly distributed in the stator circumference is a strong indicator of the magnitude of back-emf unbalance for a given slot/pole combination. However, in the case of SE, this unbalance is static, which depends on the position of the minimum airgap, and it does not change from one phase to another. Notwithstanding, no significant differences were found by comparing the results of modular and monolithic machines.

**Table 17.** Maximum back-emf unbalance for different PMSM and MPMSM slot/pole combinations in the presence of SE.

| ECC | Modular [V] | Monolithic [V] | $N_{pg}$ |
|---|---|---|---|
| 18S 12P | 0.453 | 0.527 | 6 |
| 18S 20P | 7.134 | 7.284 | 2 |
| 24S 20P | 0.391 | 0.367 | 4 |
| 24S 22P | 4.012 | 3.934 | 2 |
| 24S 28P | 0.397 | 0.539 | 4 |

### 6.5. Summary: SE and DE on Back-Emf of PMSMs and MPMSMs

In summary, it was found that eccentricity has a low impact on the back-emf of the evaluated slot/pole combinations. In this regard:

- Back-emf magnitude is not affected by eccentricity, which was observed for all the evaluated slot/pole combinations and can translate into a low impact of eccentricity on the mean torque;
- Slot/pole combinations having a high number of evenly distributed phase groups (see Figure 9) are less likely to develop back-emf unbalance. That is the case for the 18- slot, 12-pole, 24-slot, 20-pole, and 24-slot, 28-pole machines.

## 7. Results and Discussion: Mean Torque

### 7.1. DE: Comparison of Slot/Pole Combinations for PMSMs and MPMSMs

In Table 18, the mean torque for different DE magnitudes is presented. Both the modular and monolithic machines are evaluated considering different slot/pole combinations. From Table 18, it can be seen that the mean torque is not affected by DE regardless of the slot/pole combination, which was expected from the back-emf results: since the flux linkage is not considerably penalized by eccentricity, the mean torque should not be decreased either.

**Table 18.** Mean torque of different PMSM and MPMSM slot/pole combinations in the presence of DE.

| ECC $Q_s/2p$ | 0 mm (Faultless) | | 0.25 mm | | 0.50 mm | | 0.75 mm | | 1.00 mm | |
|---|---|---|---|---|---|---|---|---|---|---|
| | Mod [Nm] | Mon [Nm] | Mod [Nm] | Mon [Nm] | Mod [Nm] | Mon [Nm] | Mod [Nm] | Mon [Nm] | Mod [Nm] | Mon [Nm] |
| 18S 12P | 147.8 | 171.8 | 148.0 | 172.0 | 147.8 | 171.8 | 147.9 | 171.8 | 148.0 | 171.9 |
| 18S 20P | 218.9 | 226.1 | 218.7 | 225.8 | 218.9 | 226.0 | 218.9 | 226.0 | 219.0 | 226.2 |
| 24S 20P | 208.4 | 215.4 | 207.6 | 214.6 | 207.6 | 214.6 | 208.1 | 215.1 | 208.3 | 215.3 |
| 24S 22P | 216.2 | 218.4 | 216.2 | 218.4 | 216.3 | 218.5 | 216.4 | 218.6 | 216.6 | 218.8 |
| 24S 28P | 207.4 | 232.4 | 207.2 | 232.2 | 207.4 | 232.4 | 207.8 | 232.7 | 208.5 | 233.4 |

It could be noted that the 18-slot, 12-pole modular machine develops significantly lower mean torque than the monolithic counterpart. This can be explained by the same analysis carried out for the radial forces in Section 4.2 (see Figure 5): the absence of ferromagnetic material between stator modules penalizes the flux linkage in machines having a high value of *q*.

### 7.2. SE: Comparison of Slot/Pole Combinations for PMSMs and MPMSMs

In Table 19, the mean torque developed by the machines for different SE magnitudes is summarized, considering different slot/pole combinations. From Table 19, and complementing the findings of Table 18, it is apparent that the mean torque is not affected by DE either. The mean torque difference between machines with eccentricity and faultless machines is always lower than 0.5%.

**Table 19.** Mean torque of different PMSM and MPMSM slot/pole combinations in the presence of SE.

| ECC $Q_s/2p$ | 0 mm (Faultless) | | 0.25 mm | | 0.50 mm | | 0.75 mm | | 1.00 mm | |
|---|---|---|---|---|---|---|---|---|---|---|
| | Mod [Nm] | Mon [Nm] | Mod [Nm] | Mon [Nm] | Mod [Nm] | Mon [Nm] | Mod [Nm] | Mon [Nm] | Mod [Nm] | Mon [Nm] |
| 18S 12P | 147.7 | 171.7 | 147.7 | 171.8 | 147.7 | 171.7 | 147.7 | 171.8 | 147.8 | 171.8 |
| 18S 20P | 218.2 | 225.3 | 218.4 | 225.5 | 218.4 | 225.5 | 218.6 | 225.7 | 218.7 | 225.9 |
| 24S 20P | 207.9 | 214.9 | 208.0 | 215.0 | 208.0 | 215.0 | 208.3 | 215.3 | 208.4 | 215.3 |
| 24S 22P | 216.2 | 218.4 | 216.2 | 218.4 | 216.3 | 218.5 | 216.4 | 218.6 | 216.6 | 218.8 |
| 24S 28P | 207.3 | 232.2 | 207.2 | 232.2 | 207.4 | 232.4 | 207.8 | 232.7 | 208.1 | 232.9 |

### 7.3. Summary: SE and DE on the Mean Torque of PMSMs and MPMSMs

In summary, it was found that eccentricity has a negligible impact on the mean torque of the evaluated slot/pole combinations. Modular machines may develop a lower mean torque than monolithic machines depending on the number of slots per pole per phase.

## 8. Conclusions

In this paper, the impact of eccentric tolerances on relevant performance indices of MPMSMs was identified and described for different slot/pole combinations, as well as compared with equivalent machines with conventional monolithic stator. Both static and dynamic eccentricity were assessed for five slot/pole combinations, and the radial forces, cogging torque, back-EMF, and mean torque were analyzed. It was found that:

- Radial forces on machines with SE and DE scale with the slot number and pole number, and that the severity of these forces is lower in the case of modular machines with a high value of slots per pole per phase. In addition, a high-frequency force ripple of significant magnitude is present in machines with a slot number and pole number close to each other.
- When assessing the cogging torque, additional harmonic components (AHC) of severe magnitude are generated due to eccentricity. In this regard, machines which slot number is close to its pole number are very sensitive to eccentricity and develop a substantial cogging torque increase of up to 400%. There are no significant differences between DE and SE in terms of the cogging torque increase magnitude. Nevertheless, the harmonic order of the AHC generated by DE corresponds to the slot number and multiples, while the order of the AHC generated by SE is equivalent to pole number and its multiples.
- Back-emf magnitude and mean torque are not affected by eccentricity, indistinctly for DE or SE. This was observed for all the evaluated slot/pole combinations. However, slot/pole combinations with a low number of evenly distributed phase groups can develop a mild back-emf unbalance.

The findings of this analysis can be used in the design stage of a MPMSM and aim to enable the inclusion of the sensitiveness of each performance indicator to quickly compare the performance between slot/pole combinations. Furthermore, the observed results for the cogging torque and radial forces can be used for diagnosis ends: in MPMSMs with a slot number close to their pole number, anomalous cogging torque increases can indicate the presence of SE or DE, which can be distinguished by the harmonic order of the cogging torque signal.

**Author Contributions:** Conceptualization, D.R., C.M., and W.J.; methodology, D.R., W.J., and G.B.; software, D.R. and C.M.; validation, D.R. and C.M.; formal analysis, D.R. and W.J.; investigation, D.R.; resources, W.J., J.R., J.A.T., and G.B.; data curation, D.R. and C.M.; writing—original draft preparation, D.R., C.M., W.J., G.B., and J.R.; writing—review and editing, D.R., W.J., G.B., and J.A.T.; visualization, D.R.; supervision, W.J., J.A.T., and G.B.; project administration, W.J.; funding acquisition, W.J., J.R., and G.B. All authors have read and agreed to the published version of the manuscript.

**Funding:** This work is supported in part by the Agencia Nacional de Investigación y Desarrollo (ANID), Chile through grant ANID-PFCHA/Doctorado Nacional/2020-21200350, project FONDE-CYT REGULAR #1201667, and project FONDEF ID21I10099.

**Institutional Review Board Statement:** Not applicable.

**Informed Consent Statement:** Not applicable.

**Data Availability Statement:** Data are contained within the article.

**Conflicts of Interest:** The authors declare no conflict of interest.

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
