# Peer review of "Study on Stator-Rotor Misalignment in Modular Permanent Magnet Synchronous Machines with Different Slot/Pole Combinations"

_applsci, doi:10.3390/app13052777_

Round 1

Reviewer 1 Report

The paper is well organized. Static and dynamic eccentricity are assessed for different slot/pole combinations through the FEM, and the results are compared with those of PMSMs with conventional stator core. But, the authors need to carefully check the format of the paper for any errors, as shown in Table 8.

Author Response

The authors highly appreciate the reviewer’s comments and suggestions to improve the technical quality of the paper. A new version of the paper was prepared taking into consideration these comments, as indicated in the attached PDF file.

Reviewer 2 Report

This paper, essentially, completes recently published works of the authors in the same topic – e.g reference [25] of the paper – by providing a finite-element magnetic-field analysis-based evaluation of static/dynamic eccentricity impact on unbalanced magnetic force, cogging torque, back-emf and average electromagnetic torque for five different stator-slot/rotor-pole combinations of surface-mounted-permanent-magnet outer-rotor synchronous machines with three-phase concentrated-winding, U-shaped segmented stator. Several eccentricity magnitudes from very low to high eccentricity are considered in the study. The main findings of this evaluation of static/dynamic eccentricity effects are (i) lower unbalanced magnetic force in the case of machines with a high value of number of slots-per-pole-per-phase; (ii) marked increase of cogging torque for machines with stator-slot number close to rotor-pole number; (iii) no significant effects on back-emf and average electromagnetic torque, irrespective of stator-slot/rotor-pole combination. Although being just a research completion of elsewhere published works of the authors in the same topic, the paper may be considered self-consistent, of good quality and with pertinent results. Hence, there are no requests to be addressed to the authors. 

Author Response

The authors highly appreciate the reviewer’s comments on the first version of the manuscript. Attached to this message you may find a response letter to your comments.

Reviewer 3 Report

The article theoretically compares the effect of static and dynamic eccentricity on the performance of PMSMs with a conventional stator and with a segmented stator. PMSMs with an external rotor are considered. Motor configurations with different numbers of rotor poles and stator slots are considered. Cogging torque, back-EMF, average torque versus selected PMSM type (conventional/segmented stator), pole and slot numbers, and eccentricity are compared. In the article, some points are not clear:

1. Specify the rated power and rated speed of the motor in question, as well as in which mechanism it is planned to use it.

2. Please add an illustration with the winding diagram of the considered PMSM to the article for at least one configuration. Explain how the armature winding coil is arranged and what slot pitch it has.

3. The authors write: “In this regard, electrical machines based on permanent magnets (PMs) are well known as the most promising technology in terms of power density and efficiency [8-11], but they lack inherent fault tolerance capability”.

Explain what types of faults must the PMSM be resistant to?

4. I think that for a correct comparison, it would be necessary to carry out computer-aided optimization of best PMSM configuration under consideration, for example, using a genetic method. If designs with non-optimized parameters are compared, the results of the comparative study cannot be considered conclusive.

5. The power factor and efficiency of the PMSM configurations under consideration should also be compared.

6. At what current value are the average torques given in table 18 calculated?

7. There are no sections 6.1 and 6.2 in the text, but there is section 6.3. Please check the section numbering.

Author Response

(The authors gave the same response as above.)

Round 2

Reviewer 3 Report

I think that the authors have satisfactory addressed all my comments.